# Association of Occupational Noise Exposure and Incidence of Metabolic Syndrome in a Retrospective Cohort Study

**DOI:** 10.3390/ijerph19042209

**Published:** 2022-02-15

**Authors:** Gwansic Kim, Hanjun Kim, Byungyoon Yun, Juho Sim, Changyoung Kim, Yeonsuh Oh, Jinha Yoon, Jiho Lee

**Affiliations:** 1Department of Thoracic and Cardiovascular Surgery, Ulsan University Hospital, Ulsan 44033, Korea; 0734519@uuh.ulsan.kr; 2Department of Occupational & Environmental Medicine, Ulsan University Hospital, Ulsan 44033, Korea; 0734979@uuh.ulsan.kr; 3Department of Preventive Medicine, Yonsei University College of Medicine, Seoul 03722, Korea; yby3721@yuhs.ac (B.Y.); flyinyou@yuhs.ac (J.Y.); 4Department of Public Health, Graduate School, Yonsei University, Seoul 03722, Korea; yodasim@yuhs.ac; 5Big Data Center, Ulsan University Hospital, Ulsan 44033, Korea; fingertree@uuh.ulsan.kr; 6Environmental Health Center, University of Ulsan College of Medicine, Ulsan 44033, Korea; s2lovesky00@naver.com; 7Department of Occupational & Environmental Medicine, University of Ulsan College of Medicine, Ulsan 44033, Korea

**Keywords:** metabolic syndrome, occupational noise, cohort study, workers’ health examination

## Abstract

Metabolic syndrome is one of the common causes of cardiovascular diseases and cancers. Although noise is an environmental factor to which people can be commonly exposed at work and in daily life, there are currently insufficient studies on the relationship between noise and metabolic syndrome. Therefore, the purpose of this study is to investigate the relationship between noise and metabolic syndrome. Using a multivariate time-dependent Cox proportional hazard model, the impacts of occupational noise exposure on metabolic syndrome and its components were analyzed in a retrospective cohort of 60,727 participants from 2014 to 2017. The noise exposure group showed a significantly higher incidence of metabolic syndrome and was associated with elevated triglycerides, blood sugar, and blood pressure, but decreased high-density lipoprotein, among subgroups. There was no statistically significant association with abdominal obesity. Occupational noise exposure significantly contributed to the incidence of metabolic syndrome and changes in its components. This study could be a basis for establishing policies and guidelines to reduce noise exposure that might improve workers’ health.

## 1. Introduction

Although metabolic syndrome is defined differently for each organization, it is generally defined as a coexistence of conditions that can cause cardiovascular disease, such as insulin resistance, hypertension, and dyslipidemia. According to the definition (1998) by the World Health Organization (WHO) [1], metabolic syndrome is the co-existence of at least one of diabetes mellitus, impaired glucose tolerance, low fasting blood sugar, and insulin resistance and two or more of the symptoms of hypertension, dyslipidemia, central obesity, and microproteinuria. In addition, the European Group for the Study of Insulin Resistance (EGIR), National Cholesterol Education Program Adult Treatment Panel III (NCEP ATP III), and International Diabetes Federation (IDF) organizations also have standards for metabolic syndrome, and it is suggested that it is reasonable to use the standard proposed by 2005 NCEP ATP III criteria according to the racial standard for each country [2] (Table A1).

The prevalence of metabolic syndrome varies from country to country and is increasing in some developing countries [3], but in the United States, abdominal obesity has increased, while dyslipidemia and impaired glucose tolerance have decreased, keeping the prevalence of total metabolic syndrome constant [4]. In Korea, there was a significant increase from 1998 to 2007, but, similar to the trend in the United States, it did not change significantly until 2017. However, it seems that there is a large difference depending on lifestyle factors, as, for example, the trend of prevalence between men and women is different, and the trend of the prevalence of each diagnostic criterion is also different [5]. 

Metabolic syndrome, as shown in its definition and diagnostic criteria, is highly correlated with cardiovascular disease and diabetes, and this can lead to increased mortality [6]. In addition, metabolic syndrome can be a risk factor for not only cerebrovascular disease, but also neurological diseases such as Alzheimer’s and depression [7], and lung diseases such as asthma and pulmonary hypertension [8]. 

As there are various diagnostic criteria for metabolic syndrome, there are also multiple causative factors. First, there is westernized lifestyle. Higher caloric intake and reduced physical activity increase the risk for all factors of metabolic syndrome, and excessive calorie intake can increase the severity [9]. Genetic factors, [10], the living and working environments, and exposure to various drugs and environmental pollutants can predispose to the onset of metabolic syndrome [6]. In this manner, metabolic syndrome can result from complex interaction of various factors, and chronic exposure to diverse risk factors rather than a single event or a certain risk factor. Therefore, in order to prevent metabolic syndrome, exposure to all possible risk factors should be reduced, and thus a multifaceted approach and understanding are required. 

As it is thought that various work environmental factors are related to the occurrence of metabolic syndrome and that a lot of time is spent during the day in general in the workplace, it is necessary to study the relationship between the workplace and metabolic syndrome. One of the major chronical influencing factors of the workplace environment is noise exposure. There are studies on noise exposure and metabolic syndrome from some countries [11,12], but there are still many unknowns, including the causal relationship. Furthermore, from previous studies on the association between noise exposure and components of the diagnostic criteria for metabolic syndrome, there is a correlation between the increase in diabetes incidence and noise exposure, and prolonged noise exposure is a risk factor for insulin resistance. Additionally, prior research proposed several mechanisms of noise effect on metabolic syndrome in aspects of the central nervous system, endocrine system, and psychological stressors [13,14]. In addition, noise exposure at work increases the risk of hypertension [15] and is considered to be one of the risk factors for obesity [16]. 

This study aimed to investigate the impact of occupational noise exposure on the incidence of metabolic syndrome and on each component using a cohort dataset of workers. This study would be a motivation for polices on noise reduction in the workplace. 

## 2. Materials and Methods

### 2.1. Data Source and Subjects 

This is a retrospective cohort study using health examination data. Overall, 60,727 workers at a shipbuilding industry in Ulsan who received a workers’ health examination at Ulsan University Hospital in 2014 were selected as a research sample and observed until 2017. To evaluate the effect of noise exposure on metabolic syndrome, among a sample of 60,727 workers, there were excluded 27,162 workers who had been previously exposed to noise, 7408 workers who had already been diagnosed with metabolic syndrome, 6977 workers who failed to follow-up, and 299 workers who were diagnosed with metabolic syndrome within 1 year after the start of the study. Finally, 18,881 study subjects were selected. Among them, the noise-exposed group (*n* = 2693 subjects) and the non-noise-exposed group (*n* = 16,188 subjects) were divided and analyzed for comparison. (Figure 1).

In addition, in order to investigate the incidence rates of subgroups of metabolic syndrome according to noise exposure, subjects whose parameters exceeded the diagnostic criteria for each of the five sub-items of metabolic syndrome were excluded. Finally, 21,233 subjects with abdominal obesity, 10,834 subjects with elevated blood pressure, 15,113 subjects with elevated triglycerides (TG), 19,234 subjects with decreased high-density lipoprotein cholesterol (HDL-C), and 13,269 subjects with elevated blood sugar were selected as study subjects, respectively. The noise-exposed group and the non-noise-exposed group were divided and analyzed for comparison. (Figure A1).

This study was reviewed and approved by the Ulsan University Hospital Institutional Review Board (UUH IRB) (IRB No: 2021-06-042-001). 

### 2.2. Covariate Definition and Measurement

The subjects of this study received regular health examination every year in Ulsan, and the following items were collected from these data. Baseline demographic characteristics included age, sex, height, weight, body mass index (BMI), waist circumference, systolic blood pressure, diastolic blood pressure, smoking history, drinking history, physical activity, presence or absence of exposure to cardiovascular-related risk factors, exposure to noise, presence or absence of underlying diseases (hypertension, diabetes, and dyslipidemia) and related medication history, fasting plasma glucose level, TG level, and HDL-C levels.

Noise exposure levels at the workplace were measured with a sound meter that complies with the requirements of the American National Standards Institute. The National Institute for Occupational Safety and Health (NIOSH) reported that a time-weighted averaged recommended exposure limitation (REL) should be 85 dBA for 8 h to minimize the harmful effects of noise exposure [17], and the American Conference of Governmental Industrial Hygienists also recommended the same standard (85 dBA) [18]. Therefore, in this study subjects exposed to noise exceeding 85 dBA in the 8-hour time-weighted average at the workplace during the follow-up period were classified into the noise-exposure group, and the rest were classified into the non-noise-exposure group. 

Height (cm) and weight (kg) were measured using an automatic height weigher (GL-150 R, G Tech international, South Korea), and body mass index (BMI) was measured by weight (kg) divided by the square of height (m). According to the criteria presented by the Korean Society for the Study of Obesity [19], underweight (<18.5), normal weight (18.5–22.9), overweight (23–24.9) and obese (≥25) were classified by four stages depending on BMI. Waist circumference was measured using a tape measure at the midpoint between the highest position of the pelvis (iliac crest) and the midpoint of the lowest rib, with both feet slightly shoulder-width apart and standing in an upright position, exhaling comfortably. Blood pressure was measured using an automatic sphygmomanometer (FT-500 R PLUS, JAWON MEDICAL, South Korea) in a sitting position after resting for at least 10 min. Smoking history was classified into three groups (non-smokers, former smokers, and current smokers) according to the smoking status. Non-smokers were defined as those who had smoked fewer than 5 packs of cigarettes in their lifetime. Those who had stopped smoking were classified as ex-smokers, and those who continued to smoke were classified as current smokers. Drinking history was defined as having a drinking habit with an average of 7 or more drinks per day for men and 5 or more drinks for women at least twice a week regardless of the type of alcohol consumed [20]. Physical activity was classified into an exercise group when moderate or vigorous physical activity was performed more than twice a week, and the rest were classified into a non-exercise group. Vigorous physical activity (>6 METs) was strenuous activity that made the subject to be out of breath much more than usual (e.g., heavy lifting, digging, aerobics, or fast bicycling, etc.). Moderate physical activity (3–6 METs) was activity that made the subject to be a little bit more out of breath than usual, such as biking at a moderate pace and doubles tennis [21].

Cardiovascular-related risk factors included carbon monoxide, nitric dioxide, cyanide compounds, antimony compounds, carbon disulfide, trichlorethylene, ethylene glycol dinitrate, acetonitrile, methyl chloroform, freon 21 (dichlorofluoromethane), methylene chloride (dichloromethane), nitroglycerin, vibration, high pressure, low pressure, and exposure to night shift. This is in accordance with the Occupational Safety and Health Act of Korea. The presence or absence of cardiovascular-related risk factors was used as a covariate in the Cox proportional hazard model. All subjects fasted for at least 8 hours before undergoing the annual health examination, and then intravenous blood sampling was performed to measure fasting plasma glucose, TG, and HDL-C levels. 

### 2.3. Statistical Analysis

The general characteristics of the study subjects were expressed as mean ± standard deviation for continuous variables and frequency (%) for categorical variables. Continuous variables were compared by Student’s t-test, and categorical variables were compared by chi-square test. When studying data that have already been established over a long period of time, such as large-scale screening data, there is an immortal time bias, in other words, a guarantee-time bias, as a convenience to consider. To control this guarantee-time bias, a time-dependent Cox model and landmark analysis were performed. In the time-dependent Cox proportional risk model, the dependent variable was the occurrence of metabolic syndrome or its components, and the exposure factor was noise exposure. Sex, age, smoking history, alcohol history, physical activity level, night-shift work, and exposure to cardiovascular-related risk factors were used as the covariates. Adjusted relative hazard ratios at 95% confidence intervals were calculated using a multivariate time-dependent Cox proportional hazards model. The incidence of metabolic syndrome and 5 subgroups was marked as “event”, and the cumulative incidence rates of metabolic syndrome and five subgroups according to noise exposure were compared and analyzed using Kaplan–Meier plots, respectively. In this study, when analyzing landmarks, the landmark time was set as a time point one year after the cohort entry. All hypothesis tests were performed as a two-tailed test, and a *p* value < 0.05 was interpreted as significant. R software version 4.0.2 (R Foundation for Statistical Computing, Vienna, Austria; www.r-project.org, accessed on 7 January 2022) was used for all statistical analysis. 

## 3. Results

### 3.1. Association between Noise Exposure and Metabolic Syndrome 

#### 3.1.1. Basic Demographic Characteristics

In order to analyze the effect of noise exposure on metabolic syndrome, 18,881 subjects were finally selected from a study sample of 60,727 subjects. The basic demographic characteristics of the subjects of this study are described in Table 1.

The average age of the subjects was 41 years old, 17.8% were under 30, 31.7% were in their 30s, 24.1% were in their 40s, and 26.4% were in their 50s. Men accounted for 73.2%, and body mass index showed a distribution of 28.3% and 22.4% for overweight and obesity, respectively. Non-smokers were 51.9%, and 56% were in the non-drinking-habit group. The degree of physical activity was mostly moderately intense (86.5%). Night-shift work experience was found in 8.7%, and exposure to cardiovascular risk factors was found in 8.0%. Of the total 18,881 subjects, 2693 (14.3%) were exposed to noise, and 16,188 (85.7%) were in the non-noise-exposed group. Compared with the non-noise-exposure group, the noise-exposure group had a younger average age of 39.66 years, and the proportion of women was lower at 5.3%. In the noise-exposure group, there were more current smokers (50%), more night-work experience (18.5%), and much more exposure to cardiovascular risk factors (46.2%). (Table 1).

#### 3.1.2. Prevalence and Incidence of Metabolic Syndrome 

Among the study sample of 60,727, the prevalence of metabolic syndrome was 12.2% (7408 patients) at onset, and at an average follow-up period of 2.39 years, the number of newly developed metabolic syndrome subjects was 997 out of 18,881 subjects, showing an incidence rate of 5.3%. The comparison of the incidence rates of metabolic syndrome between the noise- and non-noise-exposure groups was statistically significant with higher incidence among the noise-exposure group (6.6%, 177/2693; 5.1%, 820/16,188, respectively, *p* < 0.001) (Figure 2).

#### 3.1.3. Analysis of Risk Factors of Noise Exposure for the Incidence of Metabolic Syndrome

Noise exposure had the crude hazard ratio of 1.48 (95% CI, 1.32–1.65) in univariate analysis, and the adjusted hazard ratio of 1.36 (95% CI, 1.19–1.57) in multivariate analysis, which showed it as a statistically significant risk factor of metabolic syndrome (Table 2). The cumulative incidence of metabolic syndrome according to noise exposure during the follow-up period was comparatively analyzed using the Kaplan–Meier plot, and the cumulative incidence of metabolic syndrome was statistically significantly higher in the noise-exposure group (*p* < 0.001) (Figure 3).

### 3.2. Correlation between Noise Exposure and Five Subgroups of Metabolic Syndrome

In the univariate Cox model, noise exposure was positively correlated with the incidence of four of the five subcategories of metabolic syndrome, and was statistically significant for: elevated blood pressure, elevated TG, lowered HDL-C, and elevated blood sugar. Noise exposure showed a negative correlation with the incidence of abdominal obesity, but was not statistically significant (Table 2). In the multivariate Cox model, sex, age, smoking history, drinking history, physical-activity level, night-shift work, and exposure to cardiovascular risk factors were used as covariates. Even in the multivariate Cox model, noise exposure was a statistically significant risk factor for the incidence of the remaining four subgroups except for abdominal obesity (Table 2).

For noise exposure, the hazard ratios for the occurrence of elevated TG and elevated blood sugar were analyzed as 1.52 (95% CI, 1.34–1.71) and 1.43 (95% CI, 1.26–1.62), respectively. The hazard ratios of noise exposure for lowered HDL-C and elevated blood pressure were 1.20 (95% CI, 1.05–1.39) and 1.18 (95% CI, 1.03–1.35), respectively (Figure 4). During the follow-up period, the cumulative incidence rates of the five subgroups of metabolic syndrome according to the noise exposure were comparatively analyzed using Kaplan–Meier plots, and it showed that elevated blood pressure (*p* < 0.001), elevated TG (*p* = 0.033), lowered HDL-C (*p* < 0.001), and elevated blood glucose (*p* < 0.001) in the noise-exposed group were statistically significant. However, no statistical significance was found for abdominal obesity (*p* = 0.12) (Figure 5).

## 4. Discussion

In this study conducted with 60,272 workers at one workplace as a study sample, the prevalence of metabolic syndrome was 12.2% (7408/60,272), and during an average follow-up period of 2.39 years, new cases of metabolic syndrome comprised 5.3% (997/18,881). In the analysis using the multivariate time-dependent Cox proportional hazard model, noise exposure was a statistically significant risk factor for metabolic syndrome with hazard ratio of 1.36 (95% CI, 1.19–1.57). In particular, it was shown that noise exposure had a statistically significant impact in elevating TG, blood sugar, and blood pressure, and lowering HDL-C among the five subgroups of metabolic syndrome, although there was no statistically significant association with the occurrence of abdominal obesity. Therefore, occupational noise exposure is a significant environmental risk factor for the occurrence of metabolic syndrome.

Although the mechanism of how noise causes metabolic syndrome has not been clearly elucidated, several possible mechanisms exist. Babisch et al. introduced two pathways of how noise affects the body: the direct pathway and the indirect pathway [22,23]. The direct pathway is activated by the immediate interaction between the auditory nerve and the central nervous system, and the indirect pathway is related to the emotional response as it represents the cognitive recognition of noise and the corresponding activation of the cortex. These two pathways cause activation of the hypothalamic–pituitary–adrenal (HPA) axis and the sympathetic–adrenal–medullary axis, which are involved in the hypothalamus, limbic system, and autonomic nervous system, leading to increase in stress hormone (cortisol, adrenaline, and noradrenaline) levels. It can trigger a variety of bodily responses, also known as stress responses. This stress response can occur through four processes: (1) autonomic nervous system and sympathetic–adrenal–medullary axis activation; (2) release of pro-inflammatory mediators, modified lipids, or phospholipids and activation of leukocyte groups; (3) vascular endothelial dysfunction due to oxidative stress; and (4) activation of the thrombus formation pathway. These pathophysiological pathways can interact with each other and can be activated acutely or chronically at various time points after exposure to noise, and can contribute to the development of metabolic syndrome and cardiovascular disease.

Until now, there have been few papers examining the relationship between noise exposure and metabolic syndrome. Recently, studies in Taiwan and the United States in 2020 covered a similar topic [11,24]. The types of noise evaluated in each study are slightly different. A Taiwanese research team studied the relationship between perceived noise and metabolic syndrome in residential areas based on the health examination data of the general public. A US-based research team studied the relationship between traffic-related noise and metabolic syndrome. This was an industrial noise study, different from other studies from Taiwan and the United States. 

A Taiwan study by Huang T et al. in 2020 analyzed the health examination data of the general public (*n* = 42,509 people), using the perceived noise-exposure data. They were estimated up to 2003 utilizing data from 2014 and 2015. Consequently, noise exposure was reported as a risk factor for the incidence of metabolic syndrome, and in particular, the higher the noise level, the higher the correlation. Furthermore, in the additional analysis of the five subgroups of metabolic syndrome, noise was significantly associated with hypertriglyceridemia, abdominal obesity, and hyperglycemia, but had no statistical significance for blood pressure and low HDL-C. Although it would be difficult to make a simple comparison because the type of noise studied was not the same, a similar result was reported, namely, that noise exposure is a significant risk factor for the incidence of metabolic syndrome, and noise was found to contribute to hypertriglyceridemia and hyperglycemia among the five subgroups of metabolic syndrome [11]. However, there were differences with our study, which found that noise exposure was also a risk factor for elevated blood pressure and lowered HDL. There may be a difference depending on the type of noise. 

In 2020, Yu et al. from the US [24], in a study of Mexican-Americans, targeted 1608 elderly (60 years or older) people to evaluate the effects of traffic-related factors (NOx and noise) on metabolic syndrome. Exposure to traffic-related noise was reported to be associated with the incidence of metabolic syndrome. This result is consistent with that of our study. Regarding the subgroups of metabolic syndrome, noise exposure was associated with hypertension, hyperglycemia, hypertriglyceridemia, and low HDL- C, but there was no statistically significant association with abdominal obesity [24]. This result is consistent with the results of our study.

Yu et al. explained that noise exposure was not related to obesity in their study because the average age of their subjects was 70 years old, an age too late for new obesity to develop during the follow-up period. They also explained that it should be taken into account that the elderly can have a disease that may reduce the abdominal circumference. Their study focused mostly on the elderly, so it is difficult to apply their findings to other age groups.

One of the reasons that noise exposure did not affect abdominal obesity despite the relatively young average age (39 years old) of the subjects in our study is that our follow-up period was relatively short. Furthermore, the fact that most of the study subjects had a high degree of physical activity may also have attenuated the effect of noise exposure on obesity. In addition, the failure to analyze all the various factors affecting abdominal obesity may also have affected our result.

The strengths of this study included the fairly large sample size, and the target of workers in a single area of industry. Furthermore, there are few studies on the relationship between industrial noise and metabolic syndrome, and it is also meaningful that this is a large cohort study on that point. However, there are several limitations of this study. First, since it is a retrospective study using large-scale screening data, there is a risk of confounding variable and immortal time biases. To overcome this, a multivariate analysis and time-dependent Cox proportional hazards model, and landmark analysis were performed. Second, one of the limitations is that the follow-up period was relatively short, with an average of 2.39 years. A long-term analysis is needed in the future. Third, it was not possible to analyze the data on air pollution and dietary habits as presented in previous papers. However, it is estimated that the exposure to air pollution was relatively similar for the subjects of this study because they worked and lived in a single area, Ulsan city. However, further analysis on this point is required in the future. Fourth, it was not possible to compare each level of noise exposure. Fifth, it was not possible to analyze the data on sitting time, which could affect the occurrence of metabolic syndrome along with noise exposure. However, all participants included here were not white-collar workers, but production workers, therefore, data on sitting time could not be included. Finally, only noise exposure in industrial workplaces was reflected, and noise exposure in residential areas where people spend time after work was not taken into account.

## 5. Conclusions

This study investigated the contribution of occupational noise exposure to the incidence of metabolic syndrome, and in particular, it demonstrated an increase in TG and blood sugar, an elevated blood pressure, and a lowered HDL-C. Managing noise exposure is an important factor in reducing the incidence of metabolic syndrome, although lifestyle factors should also be considered. Further study would be necessary to assure a comprehensive assessment of the impact of noise exposure on metabolic syndrome and subsequent cardiovascular disease risk.

## Figures and Tables

**Figure 1 ijerph-19-02209-f001:**
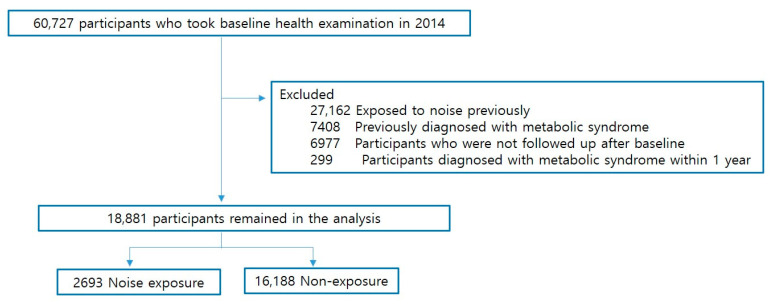
Flowchart of study subject selection to analyze effect of noise exposure on metabolic syndrome.

**Figure 2 ijerph-19-02209-f002:**
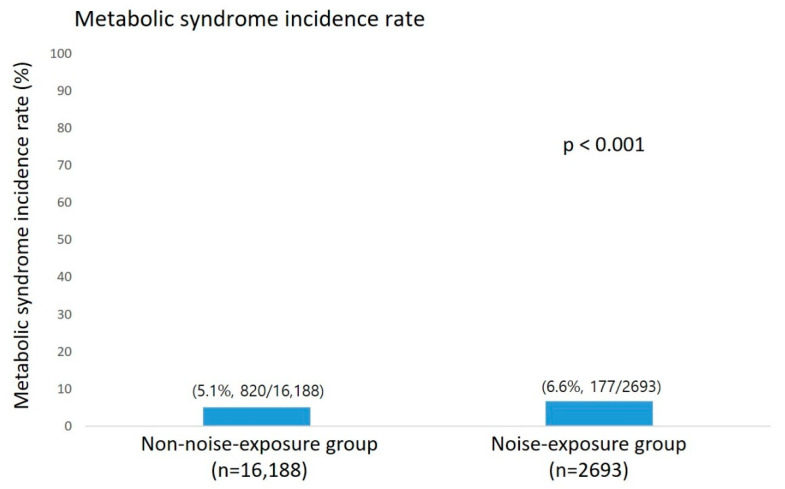
The incidence of metabolic syndrome with or without noise exposure, No. (%).

**Figure 3 ijerph-19-02209-f003:**
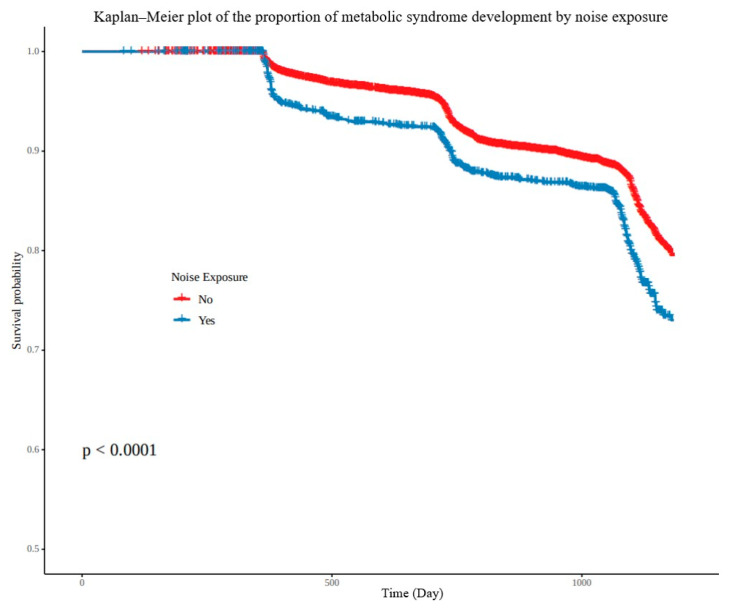
Kaplan–Meier plot of the cumulative incidence of metabolic syndrome with and without noise exposure.

**Figure 4 ijerph-19-02209-f004:**
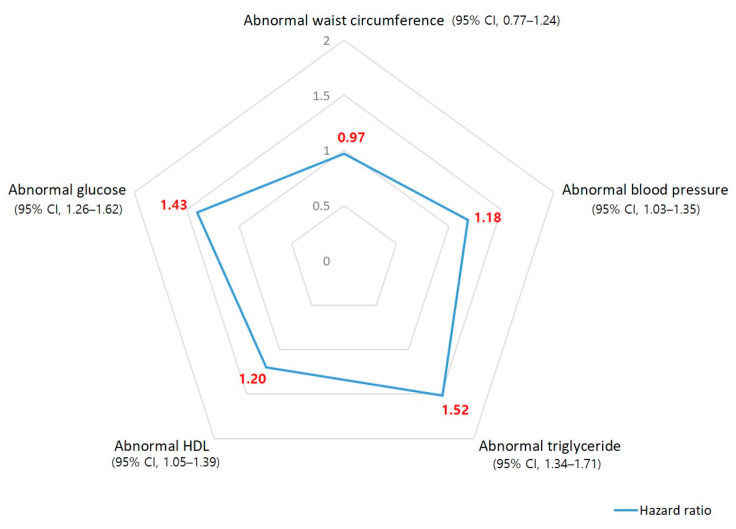
Relative hazard ratio of noise exposure for the incidence of the five subgroups of metabolic syndrome using the adjusted Cox proportional hazard model.

**Figure 5 ijerph-19-02209-f005:**
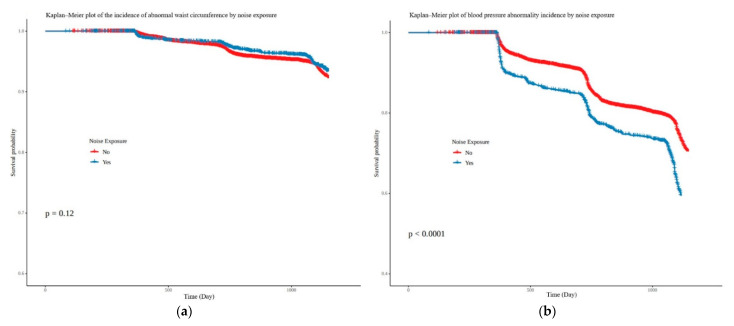
Kaplan–Meier plot of the incidence of (**a**) abnormal waist circumference, (**b**) abnormal blood pressure, (**c**) abnormal triglyceride, (**d**) abnormal HDL, (**e**) abnormal glucose by noise exposure.

**Table 1 ijerph-19-02209-t001:** Baseline characteristics of study groups for metabolic syndrome.

Characteristics	Total, No. (%) (*n* = 18,881)	Non-Exposure (*n* = 16,188)	Noise Exposure (*n* = 2693)	*p*-Value
Age (year), mean (SD)	41.04 (11.06)	41.27 (11.06)	39.66 (1.97)	<0.001
Age (year)				<0.001
<30	3369 (17.8)	2758 (17.0)	611 (22.7)	
30–39	5987 (31.7)	5174 (32.0)	813 (30.2)	
40–49	4544 (24.1)	3895 (24.1)	649 (24.1)	
≥50	4981 (26.4)	4361 (26.9)	620 (23.0)	
Gender				<0.001
Male	13,830 (73.2)	11,280 (69.7)	2550 (94.7)	
Female	5051 (26.8)	4908 (30.3)	143 (5.3)	
Body mass index, kg/m^2^				<0.001
Normal,18.5–22.9	8726 (46.2)	7498 (46.3)	1228 (45.6)	
Underweight, <18.5	589 (3.1)	550 (3.4)	39 (1.4)	
Overweight, 23–24.9	5333 (28.3)	4564 (28.2)	769 (28.6)	
Obese, ≥25	4233 (22.4)	3576 (22.1)	657 (24.4)	
Smoking status				<0.001
Non-smoker	9792 (51.9)	8972 (55.4)	820 (30.5)	
Ex-smoker	4091 (21.6)	3566 (22.0)	525 (19.5)	
Current smoker	4998 (26.5)	3650 (22.6)	1348 (50.0)	
Alcohol habit status				<0.001
No	10,581 (56.0)	9372 (57.9)	1209 (44.9)	
Yes	8300 (44.0)	6816 (42.1)	1484 (55.1)	
Physical activity				<0.001
Non-exercise group	2550 (13.5)	2105 (13.0)	445 (16.5)	
Exercise group	16,331 (86.5)	14,083 (87.0)	2248 (83.5)	
Night-shift work				<0.001
No	17,232 (92.3)	15,036 (92.9)	2196 (81.5)	
Yes	1649 (8.7)	1152 (7.1)	497 (18.5)	
Cardiovascular-related exposure				<0.001
No	17,377 (92.0)	15,928 (98.4)	1449 (53.8)	
Yes	1504 (8.0)	260 (1.6)	1244 (46.2)	

**Table 2 ijerph-19-02209-t002:** Effect estimates (and 95% CIs) from Cox models for noise exposure and the risk of metabolic syndrome for individual component.

Metabolic Syndrome and Subgroups	Univariate Model ^1^		Adjusted Model ^2^	
	HR	95% CI	HR	95% CI
Metabolic syndrome	1.48	(1.32–1.65)	1.36	(1.19–1.57)
Abnormal waist circumference	0.85	(0.70–1.04)	0.97	(0.77–1.24)
Abnormal blood pressure	1.63	(1.47–1.80)	1.18	(1.03–1.35)
Abnormal triglyceride	1.96	(1.79–2.16)	1.52	(1.34–1.71)
Abnormal HDL	1.13	(1.01–1.27)	1.20	(1.05–1.39)
Abnormal glucose	1.79	(1.62–1.97)	1.43	(1.26–1.62)

CI, confidence interval; HR, hazard ratio. ^1^ Noise exposure is the only variable. ^2^ Adjusted for baseline age, gender, smoking history, alcohol habit, physical activity, night-shift work, exposure related to cardiovascular risk.

## Data Availability

Data were obtained from health medical examination data from Ulsan University Hospital. Data sharing is not applicable.

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
