# Peer review of "Association of Occupational Noise Exposure and Incidence of Metabolic Syndrome in a Retrospective Cohort Study"

_ijerph, 2022, doi:10.3390/ijerph19042209_

Round 1
Reviewer 1 Report
- In line no. 39, there is no full form for EGIR.
- Why EGIR is not being considered for metabolic syndrome in table 1.
- In line no. 68, Reference is missing.
- In line no. 74, What kind of work environment except noise? please explain.
- In line no. 78, How noise has been defined in work environment as well as outside the work environment.
- In line no. 96, How did the author confirm that these workers were not exposed to noise?
- In line no. 108, Why 21233 ,don't it should be 1881.
- In line no. 150, Amount would be good to show what is no 7 or no 5.
- In line no. 166, All the said criteria for selection Showing tablet form would be good option.
- In Paragraph no. 3.1.2, the whole paragraph is confusing 1. The total no of included study was 18,000, 2. The language is confusing.
- In Figure no. 2, P < 0.001, Compared to what? Please explain the color indication and figure.
- In line no. 334, missing reference.
- In line no. 340, How was the association determined ?
- Major grammatical correction in article.
- In line no. 340, missing reference.
Author Response
We would like to express our sincere appreciations to the reviewers for identifying areas of corrections or modifications in our manuscript. We have made the necessary revisions in the manuscript and provided a point-by-point responses to the comments of the reviewers in the attached. The revisions in the manuscript are indicated in red.
We believe that these revisions have significantly improved our manuscript. Thank you for your consideration.

Reviewer 2 Report
Dear editor!
Thanks for the invitation to review the article.
In the paper "Association of occupational noise exposure and incidence of 2 metabolic syndrome in a retrospective cohort study" the Authors investigated the relationship between noise exposure and the incidence of metabolic syndrome. This cohort study was well designed; however I believe some points should be improved to make the paper publishable.
Comment 1:
Authors must provide information about the sample number in the abstract.
Comment 2:
In the introduction, the authors wrote: "According to the definition (1998) 34 prepared by the World Health Organization (WHO)[1], metabolic syndrome is co-exist-35 evidence of at least one of diabetes mellitus, impaired glucose tolerance, low fasting blood sugar, and insulin resistance and two or more of the symptoms of hypertension, dyslipidemia, central obesity and microproteinuria." I suggest that authors use an updated reference. This goes for all references inserted in the introduction, as the first two quotes are from more than ten years ago.
Comment 3:
The data in Table 1 should appear as supplementary material.
Comment 4:
The third paragraph of the introduction (lines 44-51) presents irrelevant information and can be removed. The introduction lacks a mechanism to justify the hypothesis that there is a relationship between noise exposure and MS.
Comment 5:
Authors must base themselves on the literature (insert citations) for the classification used in the criteria such as smokers and alcoholism.
Comment 6:
In lines 157-161, the authors describe the parameters adopted to classify cardiovascular-related risk. The description is confusing! How did the authors assess exposure to, for example, cyanide compounds, antimony compounds, carbon disulfide, trichlorethylene, ethylene glycol dinitrate, acetonitrile, methyl chloroform, freon 21 (dichlorofluoro-159 methane), and methylene chloride (dichloromethane)?
Comment 7:
It would be interesting for the authors to present the relationship between working in a sitting position and exposure to noise. This relationship is important to determine whether MS prevalence is not associated with sitting time (common in work tasks with noise exposure) or just noise exposure.
Comment 8:
In classifying the level of physical activity, the authors cite the Korean version of the International Physical Activity Questionnaire (IPAQ) short form (reference 20). However, the description for such classification does not correspond to the guidelines of the IPAQ-short form. This instrument allows to present the results in MET's and evaluates the sitting time. This last variable is relevant for the objectives of the study, as highlighted in comment 7. The authors must present a valid reference to support the classification of the level of physical activity used.
Comment 9:
Aesthetically, Figure 2 is not attractive. I suggest that the authors use a different software (eg Prism) to build the figure.
Comment 10:
The explanation of the pathophysiological mechanism of the relationship between noise exposure and MS is acceptable. However, it is interesting that the authors highlight the effect of activation of the HPA axis on the variables evaluated (glucose, HDL-c, triglyceride, and blood pressure). Furthermore, the authors should better explain (using physiological mechanisms) the reason for the negative correlation between noise exposure and the incidence of abdominal obesity. Considering the mechanism related to the activation of the HPA axis, cortisol would be expected to stimulate central obesity.
Comment 11
I suggest that the authors highlight as a limitation of the study the fact that the participants' sitting time was not evaluated. It is possible that there is a positive relationship between noise exposure and sitting time.
Author Response

(The authors gave the same response as above.)

Reviewer 3 Report
In this retrospective cohort study, Kim et al. explored the association of occupational noise exposure and the incidence of 2 metabolic syndrome. The authors concluded that occupational noise exposure contributed to the incidence of metabolic syndrome, and was associated with the increase of triglycerides, blood sugar, and blood pressure, as well as the decrease of HDL. This study is interesting and the manuscript is well desigen and written. However, several issue should be improved.
- The reason of excluding 299 workers who diagnosed with metabolic syndrome within 1 year after the start of the study should be added.
- Legends of Figure 2 should be added.
- Tables and Figures should be re-organized. The resolution is low.
- A recent work could be discussed: Diabetes Res Clin Pract . 2021 Aug;178:108944. doi: 10.1016/j.diabres.2021.108944.
Author Response

(The authors gave the same response as above.)

Reviewer 4 Report
In this study authors investigated the impact of occupational noise exposure on the incidence of metabolic syndrome. However authors did not clarify what kind of occupational noise they investigated. That information is very important for discussion of results. The nutrition of the examined is very important too.
Author Response

(The authors gave the same response as above.)

Round 2
Reviewer 1 Report
The manuscript has been revised as per the suggestion, so can be accepted for publication.